# Off-Road Drivable Area Detection: A Learning-Based Approach Exploiting LiDAR Reflection Texture Information

**Chuanchuan Zhong, Bowen Li and Tao Wu \***

College of Intelligent Science, National University of Defense Technology, Changsha 410073, China
\* Correspondence: wutao@nudt.edu.cn

**Abstract:** The detection of drivable areas in off-road scenes is a challenging problem due to the presence of unstructured class boundaries, irregular features, and dust noise. Three-dimensional LiDAR data can effectively describe the terrain features, and a bird's eye view (BEV) not only shows these features, but also retains the relative size of the environment compared to the forward viewing. In this paper, a method called LRTI, which is used for detecting drivable areas based on the texture information of LiDAR reflection data, is proposed. By using an instance segmentation network to learn the texture information, the drivable areas are obtained. Furthermore, a multi-frame fusion strategy is applied to improve the reliability of the output, and a shelter's mask of a dynamic object is added to the neural network to reduce the perceptual delay caused by multi-frame fusion. Through TensorRT quantization, LRTI achieves real-time processing on the unmanned ground vehicle (UGV). The experiments on our dataset show the robustness and adaptability of LRTI to sand dust and occluded scenes.

**Keywords:** off-road; drivable areas; BEV; texture information; multi-frame fusion; UGV

## 1. Introduction

The autonomous driving of unmanned ground vehicles (UGV) in off-road scenes has attracted more and more attention. Compared to the urban environment, the UGV will face more challenges in off-road scenes, such as a lack of high-precision maps, unstructured class boundaries, irregular features, and dust noise. In an off-road environment, the traditional methods of navigation and positioning of UGV mainly rely on the Global Positioning System (GPS). However, GPS has a certain error, and as a result, it is unreliable to only use GPS to realize autonomous navigation of UGVs in off-road scenes such as rural dirt roads and deserts, and the deviated GPS coordinates will continuously guide the UGV to the edge of the road, putting the UGV in a dangerous situation. Consequently, it is crucial to obtain drivable areas on unstructured road conditions. The local path planning of UGV relies on the detection results of the drivable areas, and the GPS locations serve as general direction guidance.

As the off-road environment is more challenging than the structured environment, the detection of drivable areas in unstructured road presents numerous challenges. Most of the drivable area detection methods use camera images as input [1–9]. These camera-based methods extract road surfaces by using geometric features such as vanishing points [10,11] or network extraction features [1]. However, the camera-based methods do not perform well due to unstructured road boundaries, dust noise, and the confusing appearance features of off-road environments. The camera is also sensitive to illumination and interfered with by smoke, rain, and snow. As a result, using image features as the main data to detect the drivable area will fail in some changeable off-road environments.

Although the point clouds of LiDAR are robust to illumination and contain 3D spatial information of the environments, the aforementioned camera problems can be partly avoided by using them to detect the drivable area. Some LiDAR-based works directly

used point clouds without pretreatment to fit the road surface [12,13] or extracted the road boundary [14–16]. They cannot work in off-road environments due to a strong dependence on the structural features of the road. With the development of deep learning methods, deep networks are used to extract effective features from point clouds. However, due to the disorder and sparseness of point clouds, the feature extraction of unstructured sources still face many challenges. Some work converts point clouds into 2D panoramic images [17] for the detection of drivable areas, but the image quality is greatly affected by the density of point clouds. Research on converting point clouds into a bird's eye view (BEV) for subsequent tasks has greatly improved [18,19]. In addition, multi-frame fusion can output more stable detection results [20]. The authors of [21] aggregated a few frames' point clouds to obtain a dense BEV height map, which is used as an input data format. When there are dynamic objects in the environment or a large pose error, the input data contains a lot of noise and error. In addition, there are few methods that take into account the dust noise and occlusion in off-road environments, as shown in Figure 1.

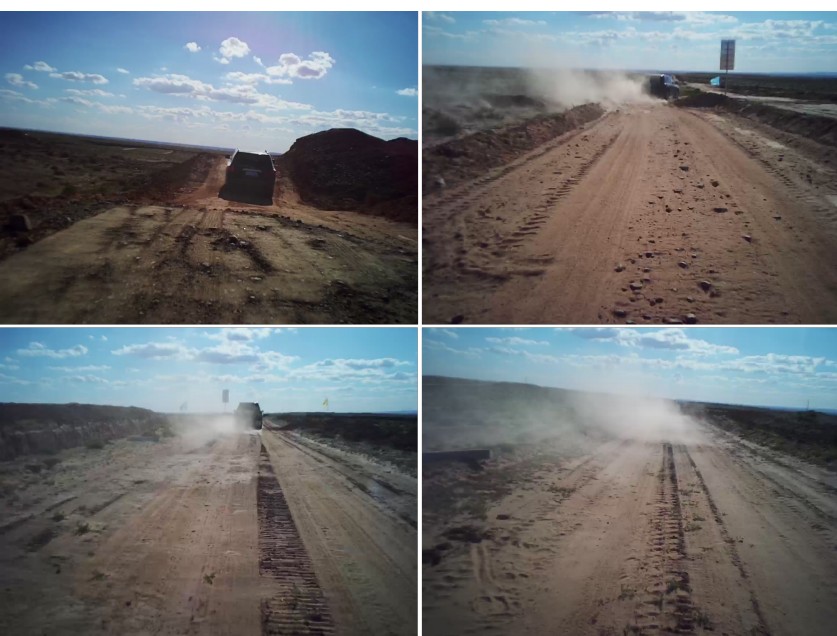

**Figure 1.** The experimental environment.

In this paper, we propose a drivable areas detection method based on the texture information of LiDAR reflection data in off-road scenes, particularly in desert environments where extracting road features was difficult. During the research, when the point cloud was projected to BEV, the texture features not only showed different ground undulations to distinguish the drivable of the area, but also displayed the rapid elevation change between the road and the non-road areas into a kind of boundary information. Then, the road model that has learned the texture features can be used to infer the drivable areas. Finally, the vehicle pose data were used to combine the detection results of continuous frames. Compared to other research in this field, the LRTI has the following benefits: first, LRTI only uses the sparse texture features of point clouds in the BEV, the algorithm can operate day and night, and can adequately handle dust scenes; second, LRTI is close to or even better than the existing methods in segmentation accuracy, as the multi-frame fusion module allows us to obtain more stable and reliable drivable areas; third, a shelter's mask of a dynamic object is added to the neural network to reduce the perceptual delay caused by multi-frame fusion when there is a large-area occlusion.

The main contribution of this paper is a complete framework for drivable area detection in off-road scenes to help UGVs plan local paths in complex and changeable unstructured environments. Our framework employs the depth network learning of the sparse texture feature of point cloud while converting to the BEV map to detect drivable

areas. In addition, we designed a multi-frame fusion strategy to achieve stable detection results. In addition to the road detection branch, a dynamic mask branch is used to solve the obstacle occlusion and the perception delay caused by multi-frame fusion when there is a large-area occlusion. The method proposed in this paper can achieve the robust detection of drivable areas in off-road environments, free from illumination restricts and dust noise.

This paper is organized as follows. First, the related works are introduced in Section 2. Section 3 introduces the methodology in detail. Section 4 presents the implementation details and experimental results. Finally, we draw conclusions in Section 5.

## 2. Related Works

The camera plays a very important role in the UGV perception system. Some camera-based methods obtain the features of a road, such as the color, the structure, and the texture, and then can further obtain potential information, such as vanishing points [10,11], road edge lines [22], and traffic lanes [23,24]. Finally, the traditional segmentation extraction method or machine learning method is used to extract the drivable areas for these features. Some works relied on the angle difference of multiple images to obtain 3D information; the authors of [25] used three cameras to reconstruct the scene and identify 3D line information contained in the scene; the authors of [26] took the parallax data measured by the stereo camera as input, analyzed each column of data in the image, and performed segmentation. In the off-road scene, the environment is complex and changeable, and the camera-based method has some bottlenecks. However, the rich and accurate 3D information of LiDAR is the solution. LiDAR and camera sensors have different features. In most cases, the fusion of the two sensors will help to better realize the detection of the driving area. Free space based on camera and LiDAR sensors [27] was compared to compute key performance indicators in real-world driving scenarios.

Due to the popularity of deep learning, various types of neural networks have been proposed, such as CNN (convolutional nueral network), RNN (recurrent neural network), and GAN (generative adversarial network). In addition, GCN (graph convolutional networks) is attracting more and more attention. It extracts features from graph data and has achieved good results in some directions [28–30]. By using deep learning, it is no longer necessary to manually design features. Compared with manual features, deep learning methods can obtain the potential features of data more easily. In application scenarios that meet specific conditions, the recognition performance of existing algorithms can be easily exceeded through deep learning; for example, in the following camera-based methods [1–9]. The smartphone camera is the most common camera in our life, and can bring many conveniences. The authors of [31–33] proposed the use of smartphone sensors to monitor road quality, identify road diseases, and locate road abnormalities. The authors of [34,35] reviewed and compared the current road anomaly detection methods using smartphones and highlighted the opportunities for further research on road anomaly detection. LiDAR–camera fusion methods have been studied in [36–40]. LiDAR is frequently used in conjunction with cameras to detect roads, and its range points are examined to estimate ground position [41] or fuse color images [42].

LiDAR-based methods have been studied in [17,20,21,43]. In [43], the proposed approach, called LoDNN, used a top-view of the point cloud as the input representation for road detection, and it used FCN [44] to segment the road. In application, the detection results of a single frame have some errors that cannot be directly output to downstream tasks. At the same time, its single frame detection accuracy needs to be improved. In [17], the authors converted unstructured 3D point clouds into 2D panoramic images containing 3D coordinates related features and used U-Net [45]/SegNet [46] to detect road areas; however, there is a problem with this input representation: the farther the pixel is, the greater the error. At a forward sensing distance of 40 meters, the error of one pixel can reach or exceed one meter in the real world. The authors of [21] aggregated a few frames' point clouds to obtain a dense BEV height map, which is used as an input data format. When there are dynamic objects in the environment or a large pose error, the input data will

have a lot of noise and error. The authors of [20] proposed that accumulating history was helpful to predict unknown regions, and it used the structure of ConvGRU to realize the pre-fusion of sparse feature map. In addition, the methods mentioned above do not take into account dust noise and occlusions. There are still many problems in the detection of dust and occlusions by the current point cloud processing methods. The authors of [47] have started to solve the problem of dust identification, but the result is not ideal. As a drivable area detection algorithm, if dust identification is required in advance, computing resource consumption will be high. This paper bypassed the recognition of dust noise and occlusions, and started from the texture information of the point cloud to realize the accurate detection of drivable areas under dust interference conditions and the cover of obstacles. Table 1 briefly summarizes the above methods.

**Table 1.** In the off-road scene, the intensity of ambient light has a great impact on the camera. LiDAR's point cloud has rich 3D structure information, which is more robust to describe the scene.

| | | | |
|---|---|---|---|
| **Traditional Method** | Camera-based | [10,11,22–26] | Low-level manual features; Weak adaptability; Poor robustness |
| | LiDAR-based | [12–16] | |
| | LiDAR-Camera | [27] | |
| **Learning Method** | Camera-based | [1–9,31–33] | Strong learning ability and adaptability; Large annotation data requirements; More computing resource requirements |
| | LiDAR-based | [17,20,21,43,47] | |
| | LiDAR-Camera | [36–42] | |

However, deep learning methods often rely on large manual annotation datasets. The authors of [28,48] proposed a semi-supervised network to reduce the dependence on labeled data. Kitti [49] and Cityscape [50], as public datasets, solve this problem to a certain extent. However, there is a lack of widely recognized and used datasets for off-road scenes. Some people work to propose relevant datasets [6,51]. In off-road scenes, camera-based methods do not perform well because image features are irregular and difficult to extract. In this scene, the more effective feature for environment description is the 3D structure feature. As a result, LiDAR data enjoy more benefits, and as an active light source, the LiDAR will not be affected by the ambient intensity. Our research is focused on the detection of drivable areas in off-road scenes. The 3D structure information of the environment is very important and the texture information in the BEV of the point cloud shows the roughness of the road.

As a result, this paper proposed a method for detecting drivable areas based on the texture information of LiDAR reflection data, called LRTI. The detection model uses the well-known mask RCNN network [52], and then LRTI fuses the detection results of the continuous frames to obtain a stable segmentation mask. When there is occlusion, it may cause perception delay caused by multi-frame fusion. As a result, a shelter class of a dynamic object is added to the model to reduce the perceptual delay; this processing also helped us attain a more realistic driving area and improved the stability of the algorithm.

### 3. Methodology

This section describes the algorithm framework and processing details. It is divided into four sections: (1) network architecture; (2) dataset preparation; (3) multi-frame fusion. The algorithm framework of LRTI is shown in Figure 2.

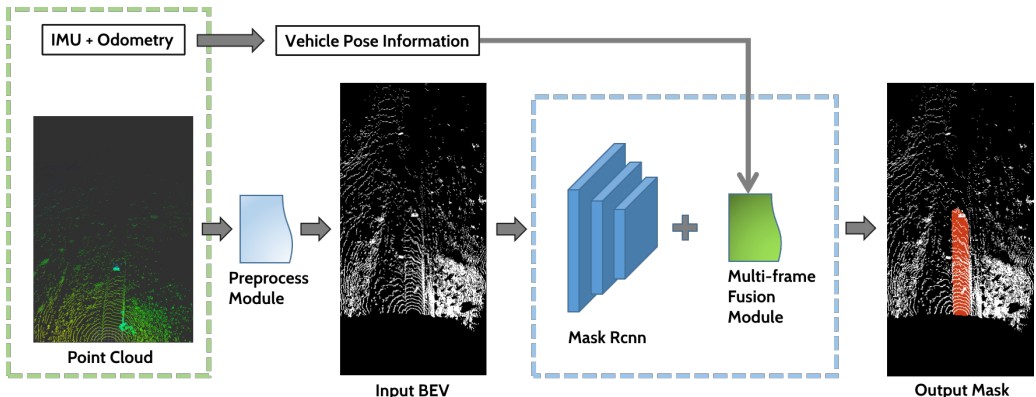

**Figure 2.** Algorithm framework. The input data are 3D point cloud, IMU data, and odometry information; through the preprocess module and some algorithms, the BEV of the point cloud and the pose of UGV are obtained; finally, the mask of the drivable area is obtained through the mask RCNN [52] network and multi-frame fusion module.

### 3.1. Network Architecture

At present, segmentation networks are classified as semantic segmentation, instance segmentation, and panoramic segmentation. The goal of semantic segmentation is to categorize all pixels in an image; instance segmentation combines object detection and semantic segmentation. In comparison to object detection's bounding box, instance segmentation can be accurate to the object's edge; different from semantic segmentation, instance segmentation requires labeling different individuals of the same category object in the image. Panoramic segmentation is a hybrid of semantic and instance segmentation. The primary distinction between panoramic segmentation and instance segmentation is that instance segmentation only processes the detected objects in the image, whereas panoramic segmentation processes all objects and backgrounds in the image. However, it is not required to classify every pixel, and in addition to detecting the drivable area, it is also required to detect the dynamic object's occluded area. Each occluded area is independent, and it is necessary to obtain the confidence of each occlusion area before post-processing. To summarize, instance segmentation is the most consistent with the desired structure of segmentation.

We chose mask RCNN [52] as the network structure for three reasons: (1) it belongs to the instance segmentation model; (2) it has relatively high mask segmentation accuracy; (3) it has good generalization performance. The network structure of mask RCNN is shown in Figure 3.

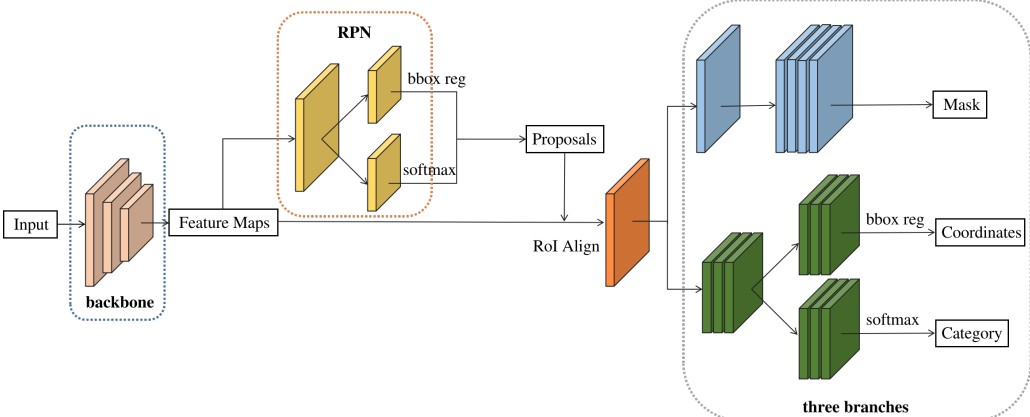

**Figure 3.** Mask RCNN.

### 3.2. Dataset Preparation

Our input representation enjoys several benefits: (1) the actual scale of the environment is maintained after the data are projected to the X-Y plane, which is fits better for the data representation of the drivable area; (2) the texture information in the BEV of the point cloud shows the undulating surface of road very well—the higher the terrain, the shorter the return distance of the point cloud for the same laser, and vice versa; (3) compared to the front view image, it is more resistant to dust noise.

We use the Robosense 80 LiDAR as our LiDAR sense, which was mounted at a height of 1.6m from the ground and operated at a rate of 10 FPS. In this work, the LiDAR was calibrated to the vehicle coordinate system, taking the right side of the vehicle as the positive direction of the X-axis, the front of the vehicle as the positive direction of the Y-axis, and the top of the vehicle as the positive direction of the Z-axis. Then, we used ROS [53] to save the data into a bag format and used the bag data during later data processing. The selection range of the point cloud in the X-axis direction was $x \in \{0.3, 80\}$ m, in the Y-axis was $y \in \{25, 25\}$ m, and in the Z-axis was $z < 1$ m. The processing in the Z-axis direction can roughly screen out the point cloud of suspended objects. The resolution of the image projected to the X-Y plane was $250 \times 500$, each pixel represents $0.2$ m $\times 0.2$ m, the center of the vehicle was located at the pixel point $[125, 400]$, and the image data were saved according to pure texture, height plus texture, reflection intensity plus texture, density plus texture and the fusion of height, reflection intensity, density and texture, as shown in Figure 4—the first four are grayscale images and the last one is an RGB image.

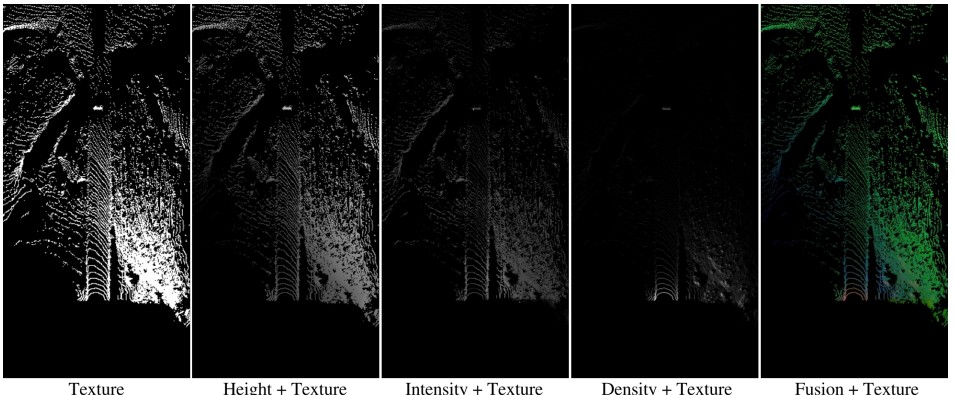

| Texture | Height + Texture | Intensity + Texture | Density + Texture | Fusion + Texture |

**Figure 4.** From left to right, the information stored in the five images is pure texture, height plus texture, reflection intensity plus texture, density plus texture, and fusion information plus texture.

In a pure texture image, if the grid of the corresponding pixel has a point cloud, it is filled in as 255. In the height map and reflection intensity map, each pixel is filled with the maximum height value and the maximum reflection intensity value of the point cloud in the corresponding grid, and then normalized to 0–255. In the density map, the value of each pixel is the number of point clouds in the corresponding grid. The farther the LiDAR reflection data points are, the more sparse they are, so only the closest reflection data can be seen from the image; the fusion map obtains the height, reflection intensity, and density information mentioned above from the three channels, respectively.

We annotated 3000 images, each image annotated the mask of the drivable area. Then, we augmented the data by flipping each image and annotation in the vertical direction. Finally, we obtained 6000 annotated images and divided them into a training set and a verification set according to the ratio of 20:1. In addition, 100 test images were prepared for collecting data in similar scenes.

### 3.3. Multi-Frame Fusion

A few frames' segmentation mask may have some errors. If the errors are output to the path planning part of the UGV, the planned path may lead the UGV to approach the edge

of the drivable area, placing the UGV into a dangerous situation. Therefore, in order to output a stable and reliable drivable area mask, it is necessary to fuse the detection results of consecutive frames.

Because LRTI is the post-fusion mask result, rather than the pre-fusion of data, there is no issue with the moving target's shadow that often occurs in multi-frame fusion. In addition, there are some errors in the vehicle pose information of each frame, and the impact of such errors should be minimized in the process of fusion.

The current vehicle coordinate system determines the coordinate system of the detection results of different frames. Consecutive detection results should be projected to the same coordinate system. The pose of the UGV can be obtained with the IMU and odometry, and we can then calculate the transformation matrix of the coordinate system at different times. The coordinate transformation matrix can be used to realize the unified representation of current detection results and historical detection results, obtaining a stable and reliable output.

Because the elevation information is reflected in the texture, only the 2D coordinate system conversion is required here. As shown in Figure 5, given a point $P'(x', y')$ in the coordinate system $X'O'Y'$ after rotation and translation, we calculated the coordinate value of $P'$ in the base coordinate system.

We found that the coordinate value of $P'$ in the coordinate system $XO'Y$. $X'O'Y'$ can be transformed into the coordinate system $XO'Y$ by rotating it counterclockwise by $\theta$, and then the coordinate system $XO'Y$ can be transformed into the coordinate system $XOY$ after translation $(-a, -b)$. In Equation (1), we show the calculation formula of the setting value of a point $P'$ in the coordinate system $X'O'Y'$ in the base coordinate system $XOY$.

$$\begin{bmatrix} x \\ y \\ 1 \end{bmatrix} = \begin{bmatrix} \cos\theta & -\sin\theta & a \\ \sin\theta & \cos\theta & b \\ 0 & 0 & 1 \end{bmatrix} \begin{bmatrix} x' \\ y' \\ 1 \end{bmatrix} \tag{1}$$

The formula for calculating the coordinate value $(x', y')$ of a point $P(x, y)$ in the coordinate system $XOY$ in the base coordinate system $X'O'Y'$ is shown in Equation (2).

$$\begin{bmatrix} x' \\ y' \\ 1 \end{bmatrix} = \begin{bmatrix} \cos\theta & \sin\theta & 0 \\ -\sin\theta & \cos\theta & 0 \\ 0 & 0 & 1 \end{bmatrix} \begin{bmatrix} 1 & 0 & -a \\ 0 & 1 & -b \\ 0 & 0 & 1 \end{bmatrix} \begin{bmatrix} x \\ y \\ 1 \end{bmatrix} \tag{2}$$

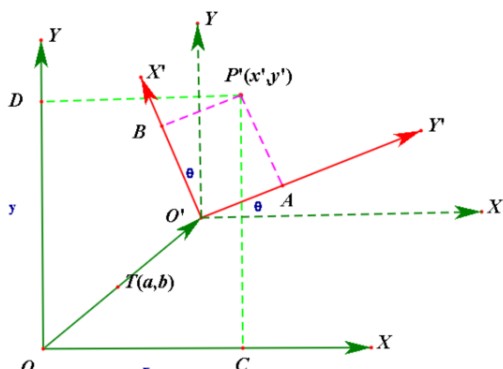

**Figure 5.** Two-dimensional coordinate system transformation.

The global coordinate system is constructed based on the initial pose of the UGV when it is started. The starting position is the coordinate origin, the front direction is positive 90° and the right-hand direction of the UGV is 0°. The pose information includes the coordinates $(dr_x, dr_y)$ of the current UGV in the global coordinate system, as well as the heading angle $dr_{heading}$ at this time. $dr_x, dr_y$, and $dr_{heading}$ is equivalent to a, b, and $\theta$ in Equations (1) and (2).

The multi-frame fusion is performed to project the mask that was used in the previous frame's coordinate system to the global coordinate system based on the previous frame's vehicle pose information. Then, we use the vehicle pose information of this frame to project the mask that was used in the global coordinate system to the coordinate system of the current frame. Suppose that the pixel value of this frame at a certain pixel position is $V_{this}$, and the pixel value of the same pixel position that the previous frame projected to this frame is $V_{last}$, $K$ is the fusion ratio, and *threshold* is the mask output threshold.

$$V_{tmp} = K \cdot V_{this} + (1 - K) \cdot V_{last}, \quad K \in [0, 1] \tag{3}$$

$$V_{final} = \begin{cases} V_{tmp} & , if \ V_{tmp} \geq threshold \\ 0 & , if \ V_{tmp} < threshold \end{cases} \tag{4}$$

The value of *threshold* is related to the mask pixel value. The value of $K$ is also related to the pixel value of the mask and the confidence of the projection mask. The error of vehicle pose information is accumulated continuously, so the longer the result is, the more unreliable it is. The $K$ value helps to reduce the impact of vehicle pose errors in the process of fusion. When the mask result of the history frame is farther away from the current frame, its contribution to the output result of the current frame is lower. Furthermore, if you are inclined to believe the detection result of the current frame, you can increase the output threshold parameter; if you are inclined to believe the mask projected to this frame, you can decrease the threshold parameter.

## 4. Experimental Results

### 4.1. Evaluation Metrics

We used the evaluation metrics of the COCO Dataset [54] in the ablation experiment. In the comparison experiment, we used some important metrics in semantic segmentation, such as class pixel accuracy (CPA), recall, intersection over union (IoU) and Dice score (Dice). The metrics here have not been averaged, because our label is background except road, and we do not care about background, so it was not added to the metric calculation.

### 4.2. Results of Ablation Experiment

We conducted four groups of experiments to validate the fitting and effectiveness of texture information, height information (with texture information), reflection intensity information (with texture information), and fusion information (including texture information, height information, reflection intensity information, and density information) in a neural network. As for the density information, it is concentrated in the area in front of the vehicle—the farther the distance is, the lower the density value is. Given that this information has a great dependence on the location, it was not listed in the experiments separately.

We selected Resnet50-FPN as the backbone of the model and used the stochastic gradient descent (SGD) as the optimizer for the training process. All our experiments were carried out on four RTX1080ti graphics cards. The learning policy adopted a linear warmup and each step learning rate was 0.01, and the batch size was 2. The maximum epoch of each experiment was 100.

The test results of the validation set are shown in Table 2, from which we can find that although other information is added, the precision is not significantly improved, and some results are not as good as the pure texture information. Moreover, the driving area belongs to the category of large area for the most part, while pure texture has higher accuracy in *mAP_l* items. From the above, it can be explained that the model mainly learns the texture features. Later, our experiments were based on pure texture information. The inference results are shown in Figures 6 and 7.

**Table 2.** Verification set test results.

| Test List | Average Precision | | | |
|---|---|---|---|---|
| | mAP bbox | mAP_s bbox | mAP_m bbox | mAP_l bbox |
| **Texture** | 0.952 | 0.920 | 0.954 | 0.969 |
| **Texture + Intensity** | 0.945 | 0.926 | 0.946 | 0.959 |
| **Texture + Height** | 0.938 | 0.920 | 0.937 | 0.962 |
| **Texture + Fusion** | 0.953 | 0.925 | 0.956 | 0.959 |
| **Test List** | **Average Precision** | | | |
| | mAP segm | mAP_s segm | mAP_m segm | mAP_l segm |
| **Texture** | 0.951 | 0.936 | 0.971 | 0.861 |
| **Texture + Intensity** | 0.947 | 0.925 | 0.968 | 0.847 |
| **Texture + Height** | 0.954 | 0.922 | 0.975 | 0.859 |
| **Texture + Fusion** | 0.948 | 0.915 | 0.966 | 0.858 |

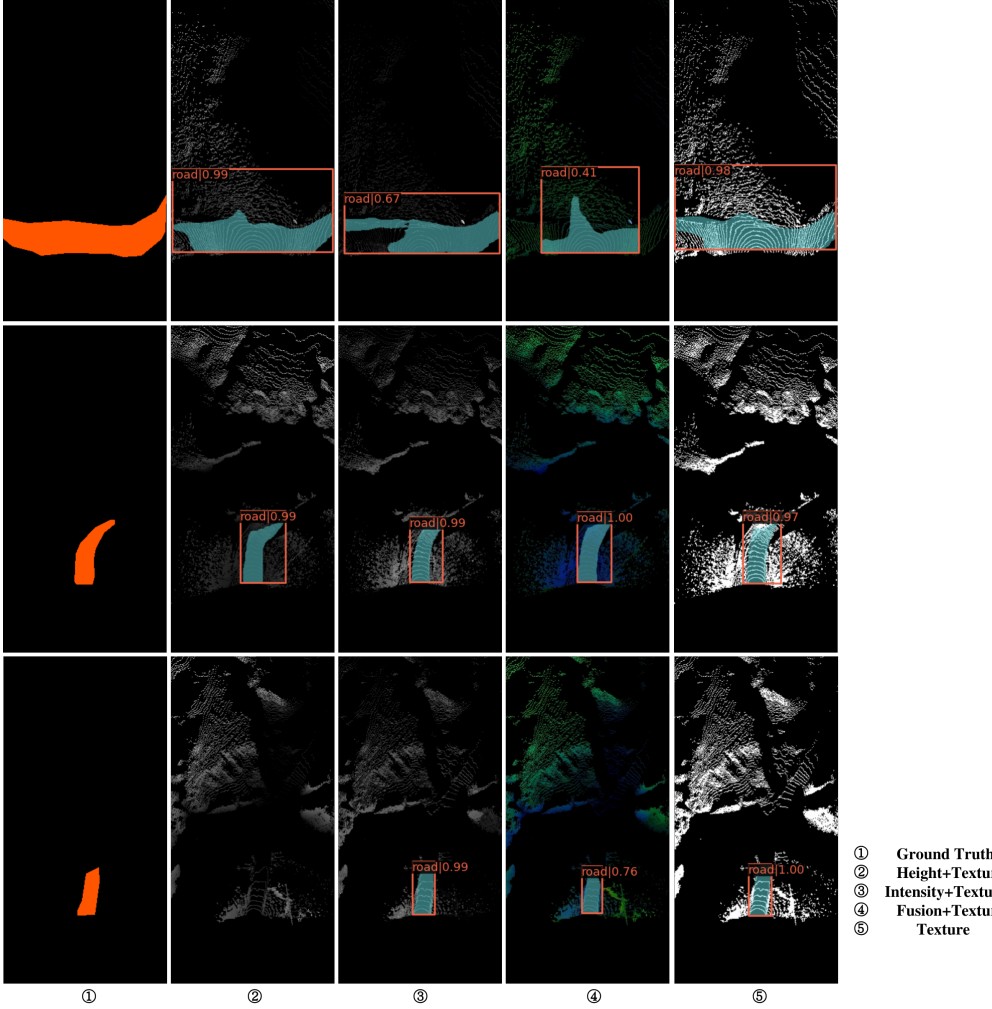

① Ground Truth
② Height+Texture
③ Intensity+Texture
④ Fusion+Texture
⑤ Texture

**Figure 6.** Display of validation set test results.

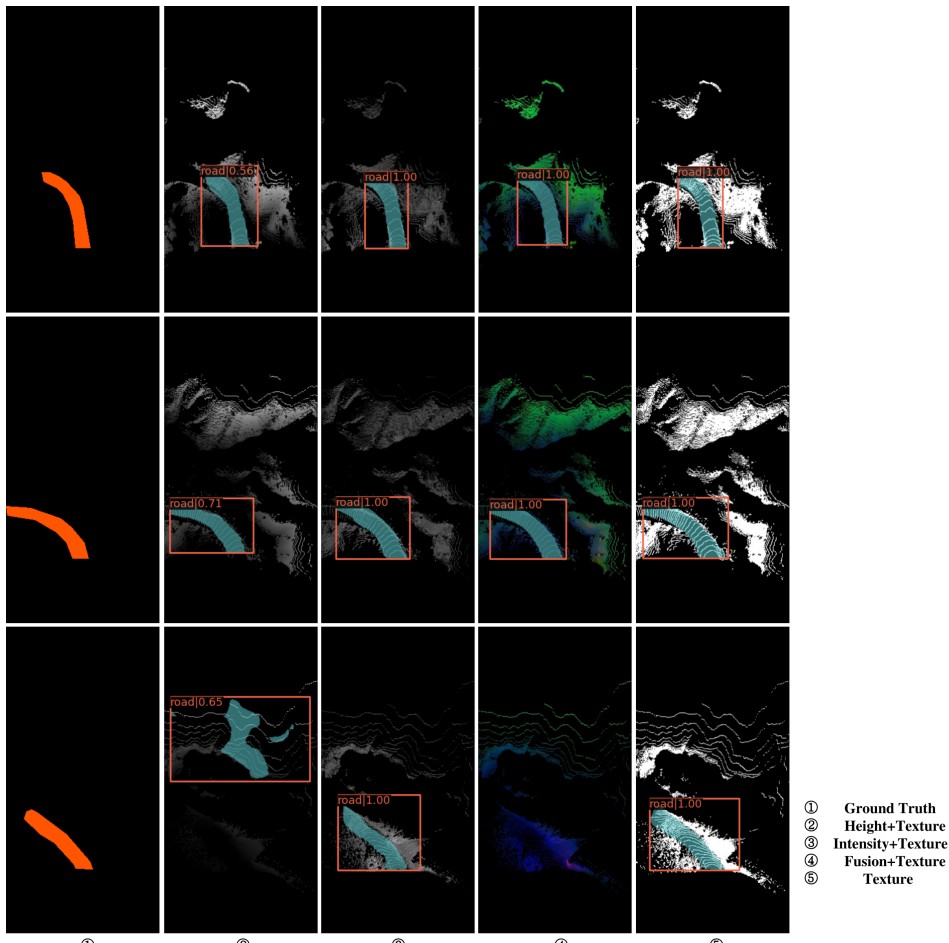

| | |
| --- | --- |
| ① | Ground Truth |
| ② | Height+Texture |
| ③ | Intensity+Texture |
| ④ | Fusion+Texture |
| ⑤ | Texture |

**Figure 7.** Display of validation set test results.

### 4.3. Multi-Frame Fusion Result

In order to obtain a stable and reliable output, LRTI fused the detection results of consecutive frames. The comparison of multi-frame fusion before and after is shown in Figure 8.

### 4.4. Comparison Results of Different Methods

We conducted a comparative experiment with LoDNN [43] represented by the same input, and the test results are shown in Figures 9 and 10. In addition, we also make a comparison with several classical point cloud segmentation networks, as shown in Table 3. From them, we could see that LRTI is better than other methods in detection accuracy. Moreover, the multi-frame fusion strategy allows our perceptual system to avoid the error of a single frame, to obtain a more stable output.

**Table 3.** Quantitative evaluation of different methods. The bold is the highest indicator of comparison methods.

| | CPA | Recall | IoU | Dice |
| --- | --- | --- | --- | --- |
| PointNet++-ssg [55] | 92.29 | - | 80.24 | - |
| PointNet++-msg [55] | 92.63 | - | 80.08 | - |
| DGCNN [29] | 92.96 | - | 81.24 | - |
| LoDNN [43] | 92.56 | 94.10 | 87.48 | 93.32 |
| LRTI (our) | **98.54** | **94.89** | **93.57** | **96.68** |

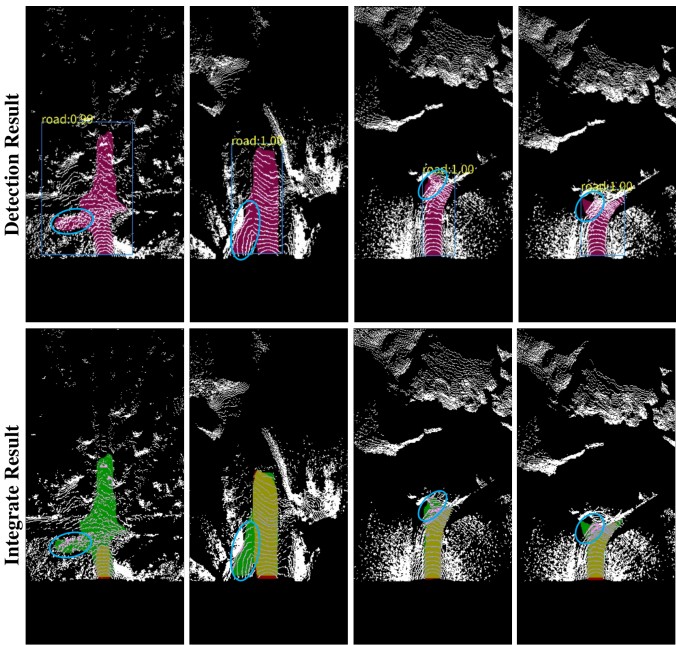

**Figure 8.** The upper row is the detection result of the original model, and the lower row is the result of multi-frame fusion. The error in the detection result of this frame is shown in the blue circle. The green plus yellow area is the detection result of this frame, and the red plus yellow area is the fusion result of this frame, which is also the final output result.

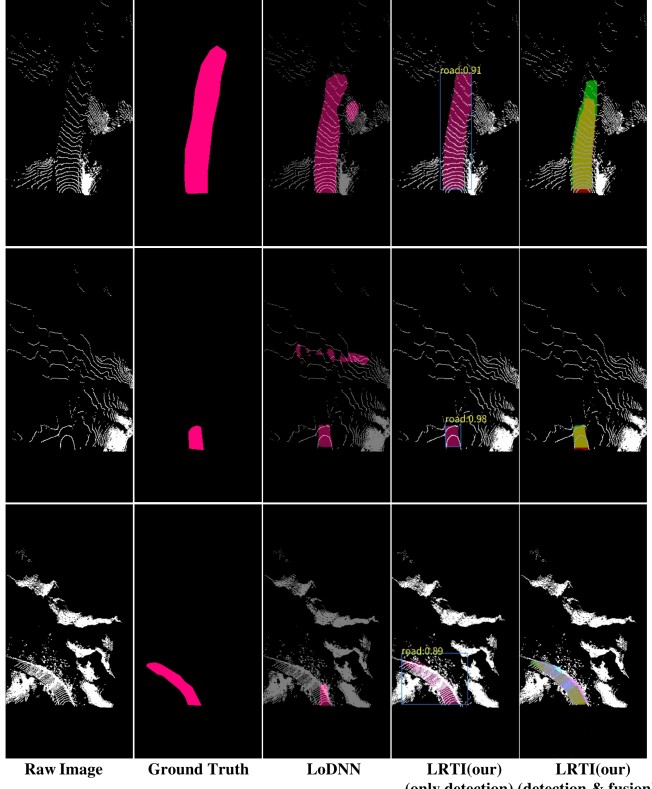

**Figure 9.** Results of different methods. Each column from left to right represents the model input image, ground truth, LoDNN [43] result, our detection result, and our detection result plus multi-frame fusion.

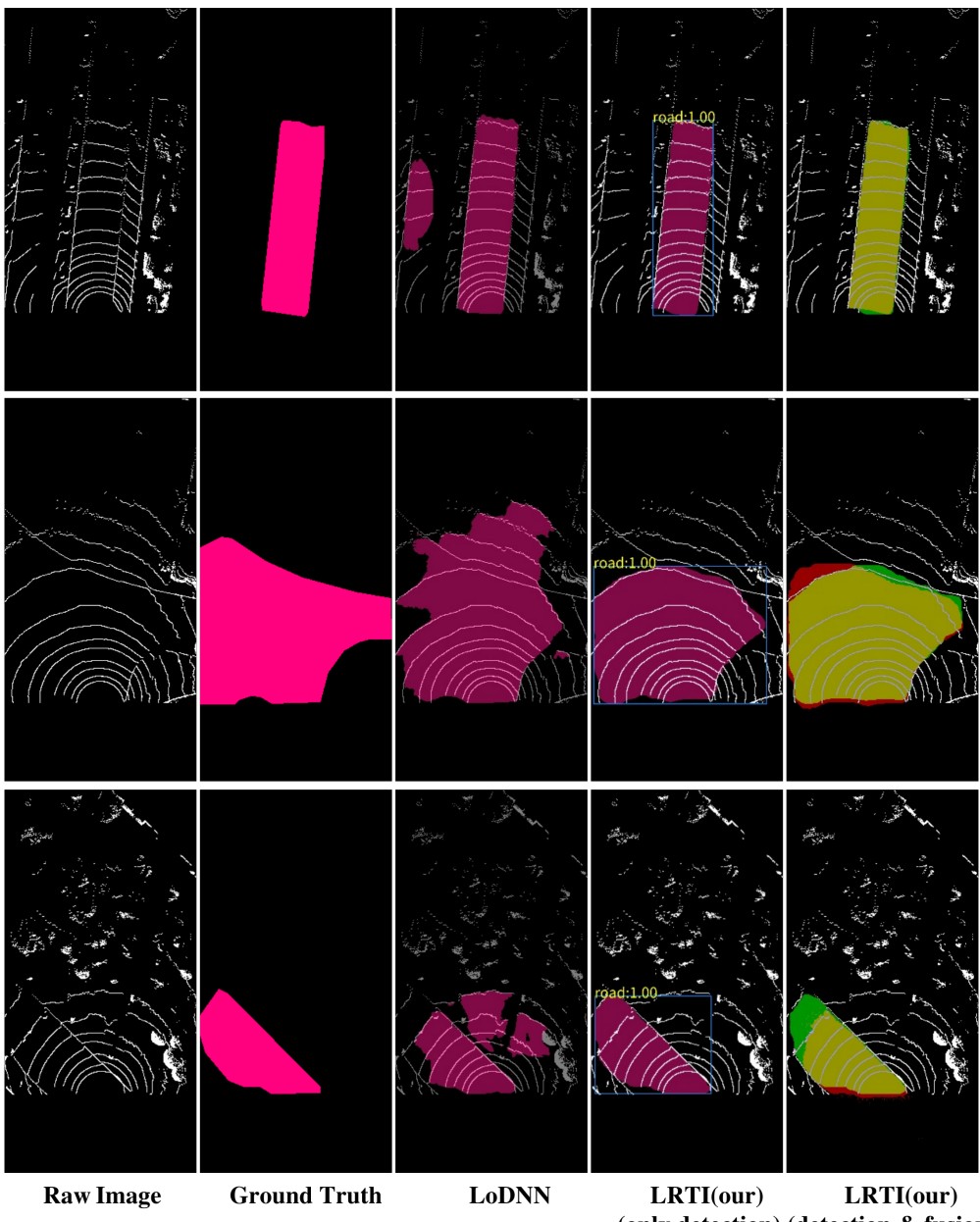

| **Raw Image** | **Ground Truth** | **LoDNN** | **LRTI(our)** | **LRTI(our)** |
|---|---|---|---|---|
| | | | **(only detection)** | **(detection & fusion)** |

**Figure 10.** Experimental results in various environments. The proposed method not only works on off-road scenarios, but also in urban roads.

### 4.5. Result of Dust Scene

In the off-road scene, the dust raised by the strong wind or the passing vehicle will have a great impact on the image and point cloud, as shown in Figures 11 and 12.

As mentioned in Section 3.2, LRTI's input representation has many benefits. Compared with the front view image, it is more resistant to dust noise under the dust interference. However, the dust is floating in the air, and most of the lasers that could have been projected to the ground can still form LiDAR reflection data; only some dense reflection data blocks of dust will appear on the BEV. During training, the influence of dust can be well treated as noise on the neural network.

In order to show the interference of dust on the detection of image drivable areas, we conducted experiments on BiSeNet V2 [56], and the experimental results are shown in Figure 12.

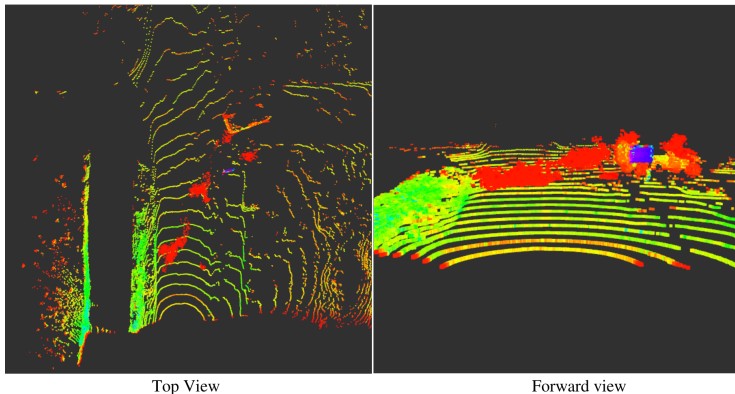

**Figure 11.** Point cloud in dust scene.

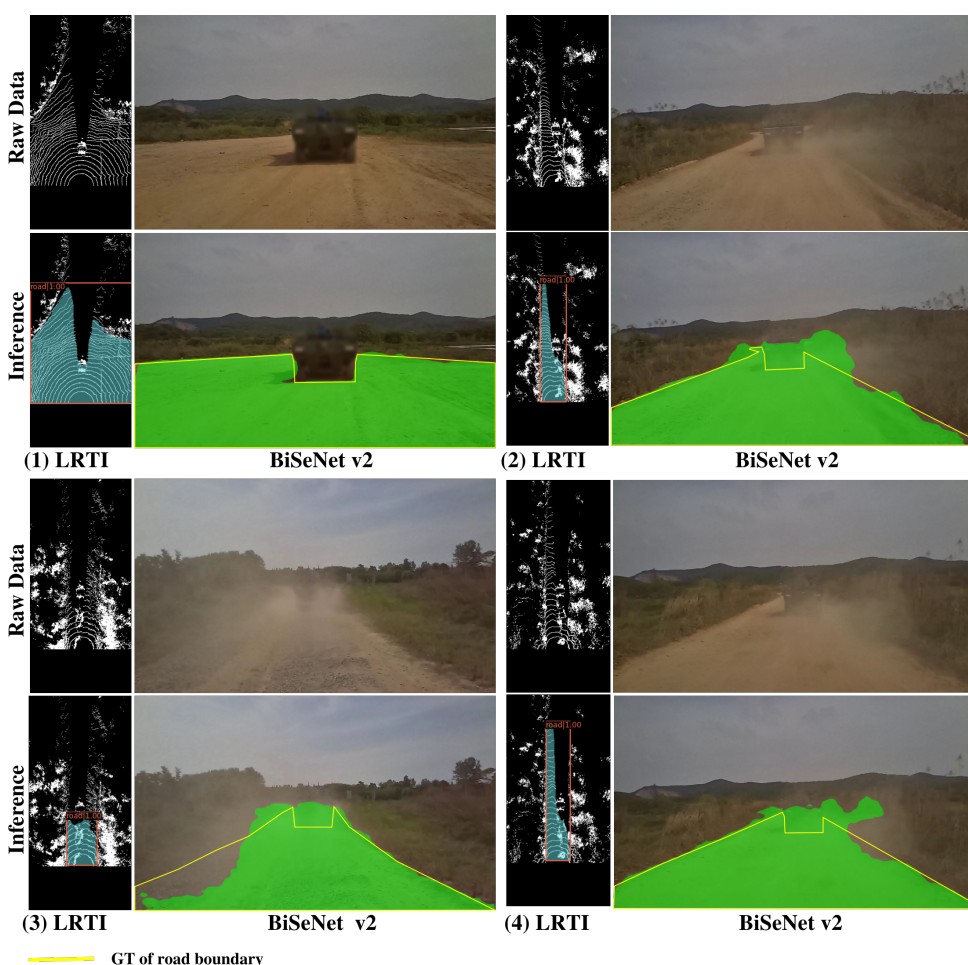

**Figure 12.** BiSeNet V2 [56] inference result in dust scenes. (1): No dust; (2)–(4): Dust.

### 4.6. Occlusion Processing Results

For the complex environment in the off-road scene, occlusion may lead to the delay of the environment perception caused by multi-frame fusion. When there is a large area of occlusion in front of the vehicle, the drivable area mask will diverge and the local path planning route will bypass the fork of the mask. Sometimes, the drivable area mask can not keep up with the speed of the local path planning, which will lead the UGV to driving to the edge of the road, which is a dangerous situation. As a result, LRTI added a mask of occlusion areas called shelter. When the road mask area detected in this frame includes the shelter mask area of the last frame, this frame would output the union area as a drivable

area mask. The results after adding the shelter mask are shown in Figure 13, and the evaluation metrics by comparing with the real mask before and after adding the shelter mask are shown in Table 4.

**Table 4.** Compare results before and after adding the shelter mask.

|                | CPA   | Recall | IoU   | Dice  |
| -------------- | ----- | ------ | ----- | ----- |
| road           | 99.03 | 83.79  | 83.11 | 90.78 |
| road + shelter | **99.14** | **94.58** | **93.14** | **96.81** |

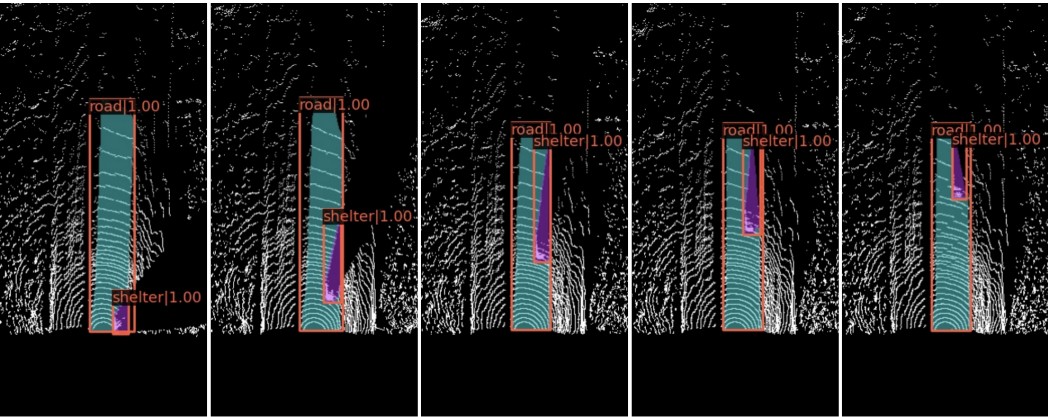

**Figure 13.** From left to right, the figure shows a car passing by our UGV.

This processing helped us to obtain a more realistic driving area and made the mask output converge quickly, reducing the perception delay caused by multi-frame fusion, and improving the stability of the algorithm.

### 4.7. Model Quantification Results

To realize the deployment on the UGV, it is necessary to achieve real-time processing efficiency. As a result, LRTI used TensorRT to quantify the model. The final experimental results are shown in Table 5. We also compare the inference speed of RTX1660ti and RTX2080ti GPUs. The results are shown in Table 6.

**Table 5.** Model quantification results.

|                                    | CPA   | Recall | IoU   | Dice  | Inference (ms) | FPS |
| ---------------------------------- | ----- | ------ | ----- | ----- | -------------- | --- |
| Raw model (RTX1660ti)              | **98.54** | **94.89** | **93.57** | **96.68** | 111            | 9   |
| FP16 quantization (RTX1660ti)      | 96.36 | 91.04  | 88.02 | 93.63 | **38**         | **26** |

**Table 6.** Comparison of reasoning speed of different GPUs.

|                               | Inference (ms) | FPS |
| ----------------------------- | -------------- | --- |
| FP16 quantization (RTX1660ti) | 38             | 26  |
| FP16 quantization (RTX2080ti) | **13**         | **76** |

Because LiDAR sends point clouds at a rate of 10FPS, it requires that the computing resources of the GPU should not exceed 100 ms when running LRTI to ensure that point clouds will not be missed, so as to achieve real-time processing.

The effect comparison before and after quantification is shown in the Figure 14. It can be seen from the figure that quantization does not bring about a significant reduction in segmentation accuracy. Under the condition of meeting deployment requirements, it is very meaningful to improve the reasoning speed in exchange for a small part of the reduced accuracy.

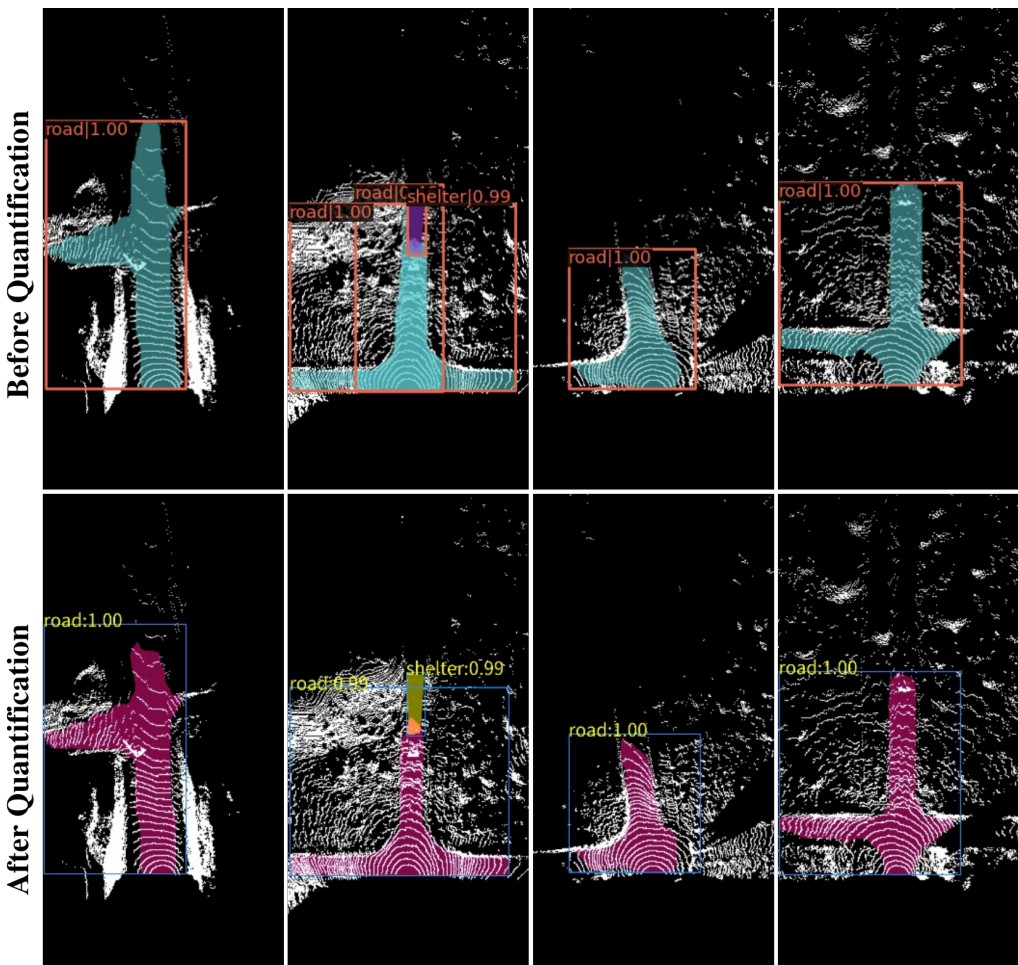

**Figure 14.** The results before and after quantification. The upper row of images is the result before quantization, and the lower row is the result after quantization.

## 5. Conclusions

In this paper, a novel approach, called LRTI, achieved the drivable area detection based on the texture information of LiDAR reflection data in the off-road scene. The texture features in BEV accurately reflect the rough ground, and LRTI applied a neural network to learn it. Furthermore, a multi-frame fusion strategy is used to improve the reliability of the output, and a shelter of a dynamic object is added to reduce the perceptual delay caused by multi-frame fusion. Through TensorRT quantization, LRTI achieved real-time processing and our algorithm system showed strong robustness and adaptability to dust and occlusion when tested with a UGV.

Limitations and future direction: The texture of LiDARs with different parameter specifications and installation methods is different, which leads to the strong dependence of LRTI on specific textures. In addition, the end-to-end training of LRTI can better extract features and reduce the accumulation of errors. In the future, we will focus on LiDAR domain adaptation and model optimization so that LRTI can be migrated to different LiDAR scans and obtain a more extensive application.

**Author Contributions:** Conceptualization, C.Z. and T.W.; methodology, C.Z. and T.W.; software, C.Z.; validation, C.Z. and B.L.; formal analysis, C.Z.; investigation, C.Z.; resources, C.Z.; data curation, C.Z.; writing—original draft preparation, C.Z.; writing—review and editing, C.Z., B.L. and T.W.; visualization, C.Z.; supervision, C.Z., B.L. and T.W.; project administration, C.Z. and T.W.; funding acquisition, C.Z. and T.W. All authors have read and agreed to the published version of the manuscript.

**Funding:** This paper is partially funded by NSFC No. 62103431.

**Data Availability Statement:** The results display and dataset can be found here: https://github.com/Zcc310/LRTI.git. Data available in a publicly accessible repository.

**Conflicts of Interest:** The authors declare that they have no known competing financial interest or personal relationships that could have appeared to influence the work reported in this paper.

## Abbreviations

The following abbreviations are used in this manuscript:

| | |
|---|---|
| CNN | Convolutional Neural Network |
| RNN | Recurrent Neural Network |
| GAN | Generative Adversarial Network |
| GCN | Graph Convolutional Networks |
| RCNN | Region Convolutional Neural Network |
| BEV | Bird's Eye View |
| 3D | Three-dimensional |
| ROS | Robot Operating System |
| GPS | Global Positioning System |
| GPU | Graphic Processing Units |
| UGV | Unmanned Ground Vehicle |

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
