# Peer review of "Off-Road Drivable Area Detection: A Learning-Based Approach Exploiting LiDAR Reflection Texture Information"

_remotesensing, doi:10.3390/rs15010027_

Round 1
Reviewer 1 Report (Previous Reviewer 2)
I would like to thank the authors for the efforts they made for improving the quality of the paper.
I recommend adding a short paragraph at the end of the introduction describing the paper's structure.
Also, the authors are invited to include a table that summarizes the content of Section 2 (about related works).
The authors also need to discuss to which extent the proposed technique can be applied in real experiments in an online fashion with real-time constraints.
The authors are invited to write a short paragraph regarding the use of smartphone sensors for detecting the status of the driver and the quality of the roads as a complement to the proposed approach. For this purpose, they may include the references below and others:
- https://ieeexplore.ieee.org/document/9327468
- https://ieeexplore.ieee.org/abstract/document/8896015
- https://ieeexplore.ieee.org/abstract/document/9089484
- https://www.mdpi.com/1424-8220/18/11/3845
-https://www.tandfonline.com/doi/abs/10.1080/10298436.2020.1809659
-https://link.springer.com/article/10.1007/s42421-022-00061-8
Author Response
Please see the attachment.

Reviewer 2 Report (Previous Reviewer 1)
In this manuscript, A learning-based Approach Exploiting LiDAR Reflection Texture Information was proposed. The idea and methodology are promising. The experimental results are also sufficient. More specifically,
1. Some background knowledges need to be further supplemented, especially deep learning.
2. The literature review is needed regarding machine learning and deep learning methods, e.g., Graph convolutional networks for hyperspectral image classification, Learnable manifold alignment (LeMA).
3. The reviewer is wondering what are the main differences between the proposed method and existing methods.
4. More experiments should be given.
5. The ablation analysis should be added as well to show the performance gain.
6. Some subsequent research directions should be pointed out.
Round 2
Reviewer 1 Report (Previous Reviewer 2)
The authors considered all my comments and suggestions. I have no more remarks to make on this paper. Good luck. Thank you for your efforts.
Reviewer 2 Report (Previous Reviewer 1)
The authors have basically fixed the comments. No more ones.
This manuscript is a resubmission of an earlier submission. The following is a list of the peer review reports and author responses from that submission.
Round 1
Reviewer 1 Report
This paper presents Off-Road Drivable Area Detection by the means of A learning-based Approach Exploiting LiDAR Reflection Texture Information. Overall, the structure of this paper is well organized, and the presentation is clear. However, there are still some crucial problems that need to be carefully addressed before a possible publication. More specifically,
1. A deep literature reviews should be given, particularly advanced and SOTA deep learning or AI methods in data processing and analysis. Therefore, the reviewer suggests discussing some related works by analyzing the following papers in the revised manuscript, e.g., 10.1109/TGRS.2020.3015157, 10.1109/TIP.2019.2893068.
2. Please clarify the contributions, why this exploration is important?
3. What are the differences in techniques between the proposed method and existing methods?
4. Some experimental results should be given.
5. Some future directions should be pointed out in the conclusion.
Reviewer 2 Report
Summary: This work proposes the LRTI approach, which uses texture information from LiDAR reflection data to identify driveable sections in off-road scenarios. The drivable zones are discovered by employing an instance segmentation network to learn the texture information. Additionally, a multi-frame fusion approach is employed to increase the output's dependability, and a dynamic object shelter's mask is introduced to the neural network to minimize the perceptual delay brought on by multi-frame fusion.
Strong points:
1. The quality of the writing in the document is high.
2. The issue at hand is significant.
3. There is a clear understanding of the issue.
4. The method that has been suggested is thoroughly discussed.
5. The results of the experiments are compelling. Detailed Comments and Suggestions: 1. When you first use the abbreviation UGV in your introduction, you should define it for the reader. 2. Always use one space before punctuation like (and [.
3. In order to provide more evidence for their statements, the writers should include more citations in the introduction.
4. The main limitations of previous work that are relevant to this research should be explained in the introduction.
5. There needs to be more emphasis placed on contributions. What is new and how it improves upon previous work should be clearly stated.
6. The authors need to be explicit about how this research differs from previous efforts and how it provides a solution. 7. A related work section is missing.
8. The authors could benefit from including a table that analyzes the primary features of similar works to show the gaps and similarities between them. The writers should think about including a line in the table to explain the workaround. 9. The following references regarding the use of smartphone sensors for detecting the status of the driver and the quality of the roads may be cited by the authors in their study:
- https://ieeexplore.ieee.org/document/9327468
- https://ieeexplore.ieee.org/abstract/document/9089484
- https://ieeexplore.ieee.org/abstract/document/8896015
- https://www.mdpi.com/1424-8220/18/11/3845
-https://journals.sagepub.com/doi/full/10.1177/0361198121990681?casa_token=uE9378SsSKAAAAAA%3AZovGvHGIuSbfEdeuBpmXspm0sNyl6rLJMcjBeMgEjeqsNpnFP84hLoaeFXoepSWmGynI14r7ZtA
-https://www.tandfonline.com/doi/abs/10.1080/10298436.2020.1809659
-https://link.springer.com/article/10.1007/s42421-022-00061-8
10. The code of the proposed solution should be made accessible on a website to ensure the reproducibility of the results.
11. Data from the experiments is not currently available to the general public. So that experiments can be replicated, authors should make it available on a publicly accessible website. 12. Short videos summarizing the results of the experiment may also be shared online. 13. The authors need to emphasize the limitations of the proposed approach in the conclusion.
14. The conclusion is too brief and could be expanded, and more future directions should be suggested.
Reviewer 3 Report
At first, my take on the paper was: Why would you use NN to handle data from this high-quality laser data source in the first place? But on the other hand, why not.
The paper converts the 3D point cloud from the laser scanner to a 2D top-view (bird's eye view) and converts the data to a 250x500 pixel black-white image, where the white pixels are pixels, where there is any detection by the LIDAR. Other data like the height and reflection intensity are converted to the same pixelized view and considered, but discarded.
After classifying road pixels and filtering using multiple frames, the road detection results seem usable. In the case of dust detection, the method seems to discard the dust area as driveable - in contrast to the vision-based method.
The detection of occulted areas, where other cars prevent detection of the road, seems like an OK method to avoid discarding driveable areas used by other vehicles. On the other hand, it is relatively easy to detect nearby moving vehicles by other means.
Objectives
I am missing a bit about the intended purpose, is a framerate of 76 FPS relevant? Is it for military use, where cost is less of an issue?
Method
The method using NN and a high-quality 3D LIDAR seems like overkill. I have seen 2D LIDAR results that can do much the same with classical methods, e.g. my own paper 'Vision Assisted Laser Scanner Navigation for Autonomous Robots' from 2008 and earlier.
I am missing some information on the sensor data, especially the type of LIDAR, its framerate and the mounting height on the vehicle.
The term 'texture' is used early in the paper without any explanation. Texture is a lot of things, and the use in this paper is simplified.
When you compare this frame with the previous, how old is the 'last frame' - in time or meters? If the framerate is 76FPS (or 26FPS), will the 'last frame' probably not provide a new perspective?
Results
Where are the limits of the method? How far away can it classify a road?
In Figures 6 and 7, how does the scene look? A small picture, like in figure 9, would be beneficial.
Can the method detect a crossing road or when the vehicle is about to enter a road from the side?
Table 1 is referenced after the table only.
Figure 8 is not referenced in the text.
The method is a good starting point if you have plenty of processing power, a high-performance LIDAR, and a short timeframe to get some results.
The method has its merits and is worthwhile reading.
Round 2
Reviewer 1 Report
The work presented a learning-based Approach Exploiting LiDAR Reflection Texture Information. Overall, the structure of this paper is well organized, and the presentation is clear. The idea is interesting and the reviewer only has few comments.
1. A deep literature reviews could be further given, particularly advanced deep learning or AI models in remote sensing. Many popular and important papers are missing. Therefore, the reviewer suggests discussing some related works by analyzing the following papers in the revised manuscript, e.g., 10.1109/TGRS.2020.3015157.
2. Some experiments should be given for comparison.
3. Some future directions should be pointed out in the conclusion.
Reviewer 2 Report
Unfortunately, the authors did not address many of my comments and suggestions. I would like to give them a second chance to deal with this.